# Thermoregulatory Response of Blackbelly Adult Ewes and Female Lambs during the Summer under Tropical Conditions in Southern Mexico

**DOI:** 10.3390/ani12141860

**Published:** 2022-07-21

**Authors:** Maricela Ruiz-Ortega, Ethel Caterina García y González, Pedro Enrique Hernández-Ruiz, Blanca Celia Pineda-Burgos, Mario Alberto Sandoval-Torres, José Vicente Velázquez-Morales, José del Carmen Rodríguez-Castillo, Elsa Lysbet Rodríguez-Castañeda, José Manuel Robles-Robles, José Luis Ponce-Covarrubias

**Affiliations:** 1Escuela Superior de Medicina Veterinaria y Zootecnia No. 3, Universidad Autónoma de Guerrero (UAGro), Tecpan de Galeana 40900, Guerrero, Mexico; maricela_ruiz@uaeh.edu.mx (M.R.-O.); 17905@uagro.mx (E.C.G.y.G.); 17688@uagro.mx (P.E.H.-R.); 13010@uagro.mx (B.C.P.-B.); 2Instituto de Ciencias Agropecuarias, Universidad Autónoma del Estado de Hidalgo, Tulancingo de Bravo 43600, Hidalgo, Mexico; 3Centro de Bachillerato Tecnológico Agropecuario No. 127, Tomatlan 48450, Jalisco, Mexico; sandovaltorres504@gmail.com; 4Instituto Mexicano del Seguro Social, Órgano de Operación Administrativa Desconcentrado, Hospital General de Zona No. 57, Cuautitlan 54769, Estado de Mexico, Mexico; civentico@gmail.com; 5Facultad de Medicina Veterinaria y Zootecnia, Benemérita Universidad Autónoma de Puebla, El Salado, Tecamachalco 72570, Puebla, Mexico; jose.rodriguez@correo.buap.mx (J.d.C.R.-C.); elsa.rodriguez@correo.buap.mx (E.L.R.-C.); manuelrobles@correo.buap.mx (J.M.R.-R.)

**Keywords:** female lambs, Blackbelly ewes, heat stress, physiological variables

## Abstract

**Simple Summary:**

Climate change has intensified high environmental temperatures, subjecting farm animals to heat stress conditions. In this study, the thermoregulatory responses of Blackbelly adult ewes and female lambs, during the summer under tropical conditions, in southern Mexico were analyzed. The combination of temperature and relative humidity in summer in tropical conditions causes heat stress in hair ewes, resulting in thermoregulatory problems. To counteract these effects, females modify their physiological constants to avoid reproductive and productive problems under these production conditions.

**Abstract:**

High environmental temperatures cause heat stress in ewes, resulting in thermoregulatory problems. In this study, the thermoregulatory responses of Blackbelly adult ewes (G1, *n* = 14) and female lambs (G2, *n* = 7), during the summer under tropical conditions, in southern Mexico were analyzed. Different physiological variables and skin temperatures (ST) of the ewes were recorded. Breathing frequency (BF) values were similar between groups at 116.73 ± 33.598 bpm (G1) and 113.661 ± 34.515 bpm (G2) (*p* > 0.05). In the case of skin elasticity (SE), there were no significant differences between the time of day and the age of the ewes (*p* > 0.05). Significant differences were observed between groups for BF, rectal temperature (RT), and heart rate (HR) values (*p* < 0.05). All ST values, for both groups, were significantly higher during the afternoon (*p* < 0.001). In general, all Blackbelly adult ewes and female lambs during the summer present severe heat stress conditions as a result of an increase in physiological constants and ST. It is concluded that all ewes thermoregulate body temperature by modifying different physiological variables to counteract the effect of heat stress.

## 1. Introduction

In recent years, a phenomenon known as global warming has been observed worldwide, derived from greenhouse gas emissions, which has created situations that threaten animal production systems and consequently food security [1]. Climate change has caused unpredictable variations in the time and amount of rainfall, as well as an increase in environmental temperature-humidity index (THI) values [2]. These problems favor the presence of heat stress conditions in domestic animals, causing health issues, reduced production, and an increase in the percentage of mortality [3].

Mexico has various agroecological regions that range from deserts to tropics, the latter representing 30% of the country [4,5]. In tropical regions, high ambient temperatures (35 °C) and maximum relative humidity (75%) represent a challenge for ewes under heat stress [6,7]. Indeed, it has been observed that, during the summer, ewes increase their BF, RT, HR, and ST in body areas such as their head, neck, scapula, stomach, and udder [8,9]. In ewes, these physiological variables increase when the THI increases to >72 units [10], however, Marai et al. [11] reported that heat stress began at 82 units (classified as moderate heat stress (from 82 to <84 units), severe (from ≥84 to <86 units) and very severe (≥86 units)]. This has been reflected in thermoregulatory problems of ewes and serious consequences for reproduction, gestation, and production [2,12].

In the country, ewes have shown heat stress tolerance to climatic conditions in regions with high ambient temperatures [13]. The breed of ewes that has shown the best adaption to heat stress conditions is the Blackbelly ewe, originally from Africa [14]; this breed has high hardiness and prolificacy in humid and sub-humid environments such as those found in the tropics, and therefore, it has been well managed in southeast Mexico [15]. It has been shown that hair ewe breeds have the capacity to grow and reproduce under heat stress conditions due to the fact that they have thermo-tolerance genes; in addition, phenotypically, the structure of their skin and hair gives them advantages over other wool breeds to dissipate body heat efficiently by evaporative and non-evaporative pathways [16,17]. These animals have the ability to activate physiological, metabolic, and endocrinological mechanisms, which help to maintain adequate body water balance and homothermic conditions at environmental temperatures from 38.3 to 39.9 °C, at a low energy cost [8,18].

In addition to temperature and relative humidity, solar radiation together with low wind speed increase an animal’s heat load, resulting in poor performance, decreased animal comfort, and death [19]. Although THI values have been used effectively as indicators of heat stress, an adjustment to the THI has been recommended that includes two additional factors, i.e., radiation and wind speed. These two factors significantly influence heat load; in addition, changes in wind speed result in convective cooling which helps to decrease heat stress [20,21], which is reflected in thermoregulatory problems with maintaining a comfort zone in ewes and serious consequences for reproduction, gestation, and production [2,11].

Some studies have reported that hair ewes could lose up to 60% of their heat through the respiratory tract [8,22]. Da Silva et al. [23] stated that decreased BF was related to the loss of heat by convection through ewes skin. Infrared thermography through cameras and pistol-type infrared thermometers are useful tools to measure ST in production animals [24,25], therefore, making it possible to reduce the use of invasive techniques such as the use of a lubricated digital thermometer inserted into the rectum to record RT [26]. The foregoing studies have demonstrated that technology could work as a tool to measure the effect of heat stress on ewes in warm environments [2,24]. Therefore, the aim of the present experiment is to analyze the thermoregulatory responses of Blackbelly adult ewes and female lambs, during summer under tropical conditions, in southern Mexico; we hypothesize that adult ewes have greater thermoregulation than female lambs under heat stress conditions.

## 2. Materials and Methods

### 2.1. Experimental Conditions

The present experiment was carried out during the months from May to July 2020 at the Ewe and Goat Zootechnical Post of the Higher School of Veterinary Medicine and Zootechnics No. 3 “Campus Costa Grande”, UAGro, located in the municipality of Tecpan de Galeana, Guerrero, Mexico. The region is located in the tropics at 17°06′57″ N latitude and 17°41′33″ W longitude with respect to the Greenwich meridian. The characteristic local climate is classified as a semi-warm sub-humid environment, with maximum ambient temperatures (40 °C) during the summer and minimum temperatures (17 °C) during the winter [6]. All animal management procedures were conducted within the guidelines of national approved techniques for animal use and care [27]. The experimental protocol was approved by the Use and Care of the Animals in Experimentation Committee of the Universidad Autonoma de Guerrero (protocol #098).

### 2.2. Animals and Treatments

In the study, twenty-one Blackbelly ewes were used, which were divided into two groups (G1 and G2): G1 consisted of fourteen multiparous empty adult ewes aged 2.79 ± 0.94 and G2 consisted of seven female lambs of four months of age. At the beginning of the study, the adult ewes had an average live weight of 35.32 ± 1.41 and 19.27 ± 1.25 kg and body condition (BC) scores of 2.6 ± 0.4 and 2.4 ± 0.2 units, respectively, measured on a 5-point scale, from 1 = emaciated to 5 = fat [28] (Table 1).

### 2.3. Climatic Variables

The climatic data were requested via email from the National Meteorological Service [6], specifically from the Atoyac weather station (DGE), number 12161, located 1 km from the study site [29]. The station recorded the following information: wind speed (km/h), environmental temperature (°C), relative humidity (%), and solar radiation (W/m^2^). The data provided by the station were recorded every 10 minutes, 24 h during the months of May, June, and July 2020. With the above information, the temperature-humidity index (THI) was calculated with the equation proposed for cattle [30] as follows:THI = T − {[0.55 × (1 − RH)] × (T − 14.4)}(1)
where T represents the ambient temperature and RH the relative humidity in decimals.

In addition, the calculation of the temperature-humidity-wind-radiation index (THWRI), according to Mader [19], was incorporated as follows: THWRI = [4.51 + THI − (1.992 × WS) + (0.0079 × SR)]
where WS represents wind speed in km per hour and SR represents solar radiation in units of watts per square meter.

THI and THWRI were calculated for three hours of the day: morning (from 6:00 am to 12:00 pm), afternoon (from 12:10 pm to 6:00 pm), and night (from 6:10 pm to 5:50 am).

### 2.4. Physiological Variables and Skin Temperatures

The physiological variables evaluated were: breathing frequency (BF), heart rate (HR), rectal temperature in Celsius degrees (RT), skin elasticity (SE), capillary return time (CRT), and ruminal movements (RM); SE and CRT were measured in seconds, the other variables were measured for one minute [31]. BF was measured by counting the number of movements of the right paralumbar fossa; HR was measured with a single bell stethoscope (3M™ Littmann^®^, Penlight; Shanghai, China), counting the number of repetitions per minute; RT was measured with a digital thermometer of veterinary use (Delta Trak^®^, Pleasanton, CA, USA). In addition, the following skin temperatures were measured: head, scapula, right paralumbar fossa (RPF), haunch, leg, and abdomen [2]. For the skin temperatures (ST), full body photos of each ewe were taken from a distance of 2.5 m with an infrared thermography camera (Fluke Ti10, Everett, WA, USA). To view photos with the Fluke Smart View^®^ 3.9 software they were downloaded to a computer. All variables were recorded in each experimental unit at 7:00 am and 3:00 pm.

### 2.5. Accommodation and Feeding

The two groups of animals were housed in an open pen built with wooden posts, chain link mesh, and a galvanized roofed galley (13.20 m long and 13.80 m wide). All the feeders were made of wooden boards (2.10 m long and 0.39 m wide, the second 1.14 m wide and 38 m wide), and plastic tub drinkers (0.96 m wide and 0.68 m high). All ewes remained grazing in the morning (9:00 am) and afternoon (5:00 pm), consuming guinea grass (*Panicum maximum*) and grass (*Cynodon dactylon*). The corral always had clean and fresh water freely accessible that the animals consumed when they returned from grazing.

### 2.6. Statistical Analysis

The data were subjected to an analysis of variance under a 2 × 2 factorial design with repeated measurements over time using the SAS statistical program [32]. Initially, the data were analyzed using descriptive statistics: calculating means, standard deviations, coefficients of variation (minimum and maximum values) for each climatic and study variable using the PROC MEANS option. The means were separated with the PDIFF command at a significance level of 5%. The means of the variables studied were compared with the Tukey test.

## 3. Results

### 3.1. Weather Conditions

The environmental conditions recorded during the experiment are presented in Table 2. The mean temperatures during the experiment were 29.3, 34.2, and 28.5 °C, during the morning, afternoon and night, respectively. The maximum environmental temperature recorded was 38.8 °C in the afternoon and the minimum temperature was 24.7 °C in the morning. The average relative humidity values were 70% in the morning and in the afternoon, and 78% at night, with registered maximums of 90% (Table 2). During the study, mean and standard deviation (SD) wind speeds of 4.5 ± 1.6, 6.1 ± 2.1, and 4.22 ± 1.72 km/h were observed in the morning, afternoon, and evening, respectively. Regarding solar radiation, the highest average was recorded during the afternoon (561.7 ± 346.7 W/m^2^).

The THI results ranged between 77 and 90 units (U); THI values of 79 U were obtained during the morning, 88 U during the afternoon, and 77 U at night (Figure 1). THWRI values recorded were 79 U during the morning, 88 U during the afternoon, and 77 U at night, with a maximum of 88 U.

### 3.2. Physiological Variables

In the present investigation, on the one hand, no significant differences were found in the BF of G1 adult ewes and G2 female lambs (*p* > 0.05). On the other hand, statistical differences were recorded in the BF by time of day (morning vs. afternoon) (*p* > 0.05). BF was higher in the G1 adult ewes during the afternoon (minimum (116.73 rpm) and maximum (190 rpm)) (Table 3). Likewise, RT was higher in the G2 female lambs during the afternoon (39.26 °C). The G1 adult ewes had a higher HR in the afternoon (108.5 bpm); however, during the morning, HR was similar between the two groups (*p* > 0.05).

On the other hand, the SE variable did not present significant differences between groups of ewes (*p* > 0.05). In the case of CRT, the highest value was presented during the afternoon for both groups of ewes (*p* < 0.006). RM were also higher during the afternoon 1.39 ± 0.48 pdm (G1) and 1.23 ± 0.42 pdm (G2) (*p* < 0.001) (Table 4).

### 3.3. Skin Temperatures

The ST results indicate that all temperatures are statistically different by time of day (*p* < 0.001) (Table 5). The highest ST was recorded in the haunch area (42.01 ± 6.93 °C) in the G1 ewes during the afternoon (*p* < 0.001).

## 4. Discussion

The environmental conditions recorded in this study (THI values from 77 to 89 U) are indicative of severe heat stress in ewes under tropical conditions. The environmental temperatures (maximum 38.8 °C) and relative humidity (maximum 90%) recorded during the experiment were higher during the afternoon. The physiological constants, CRT and RM, were higher during the afternoon regardless of the group. Finally, the highest ST was recorded in the anatomical region of the haunch in G1 during the afternoon.

### 4.1. Weather Conditions

The thermoneutral zone for ewes has been identified to be between 12 and 27 °C [1,11], the upper limit of this zone is considered to be 30 °C [33]. In the present study, Blackbelly ewes remained outside of the thermoneutral zone during the recorded hours of three times periods each day (morning, afternoon, and night) in the municipality of Tecpan de Galeana, Guerrero, Mexico.

Combinations between temperature and relative humidity with THI values ≥ 78 U indicated sufficient environmental conditions to produce heat stress in ewes [30]. Neves et al. [33] found that hair ewes began to show signs of heat stress when the THI value reached between 78 and 79 U. Hair ewes have greater tolerance to heat stress conditions because of genetic and phenotypic adaptations, as well as the activation of physiological, metabolic, and endocrinological mechanisms, which help to maintain adequate body water balance and normothermic conditions [13].

During the recorded hours of the three time periods each day (morning, afternoon, and night), the Blackbelly ewes, in the present study, remained under heat stress classified as severe. This result coincided with that reported by Macías-Cruz et al. [34] during the month of June in lambs (THI values of 60.4 and 78.5 U), however, the RH conditions were different in the geographical areas, since the RH was less than 55% in the Mexicali Valley, State of Baja California, in northwestern Mexico. Hair ewes tolerate higher temperatures than wool breeds, therefore, it is expected that the THI where any breed begins to experience signs of heat stress is <79 U [13]. The environmental temperatures at night were slightly lower (28.5 °C), which was reflected by the THI value during the afternoon (88 U) and night (77 U).

Taking the study by Arias et al. [35] as a reference, it is suggested that when the THI values at different temperatures and relative humidity are adjusted with wind speed and solar radiation under conditions of 5 m/s and 250 W/m^2^, the THI is considered to be a dangerous environmental situation with 79 < THI ≥ 84.

Wind can help to reduce the effects of heat stress during the summer by improving heat dissipation processes, mainly through evaporation pathways [36]. Wind flow is considered to be turbulent at speeds greater than 60 km/h and, when its speed is very low (<1 km/h), it is classified as a light breeze. Wind speeds greater than 5 km/h have been shown to facilitate heat transfer in ewes [37]; in this study, this phenomenon could have occurred during the afternoon when maximum speeds of 19.5 km/h were reached. The presence of hair or wool on the surface of the animal is an effective means of reducing heat loss or gain by forced convection by reducing the effect of wind on energy exchange; short-haired ewes maintain efficient heat removal by this mechanism.

As compared with the means of solar radiation of 800 W/m^2^ in the months of May, June, and July in the Yaqui Valley, Sonora, this research was at lower levels than those reported in [38]. Direct and indirect solar radiation is considered to be one of the most important factors that affects thermal balance in cattle [39]. Other investigations have shown that solar radiation has a direct impact on RT and BF [38,40]. Although the THWRI incorporates climatic variables such as wind speed and solar radiation, we did not find studies where the heat stress level in ewes was categorized using THWRI. Taking the THI as a reference and the climatic conditions of the region, we propose severe heat stress for hair ewes at THWRI values greater than or equal to 75 U. It is necessary to take the THI and the THWRI values into consideration to determine heat stress in ewes from hot and humid regions such as those found in the tropics of Guerrero.

### 4.2. Physiological Variables

In ewes, the normal values of the physiological constants vary according to the age and size of animals, which, on average, RTs between 39 and 39.6 °C and BFs between 90–100 beats per minute (bpm) and 10–20 bpm [41]. As compared with the study by Meza-Herrera et al. [42], where they reported BF greater than 160 bpm at a THI value of 79.1 U, the Blackbelly ewes in this study were under heat stress similar to the previous study, because BFs greater than 180 bpm were recorded in the afternoon in the ewes. This adaptive response allows the animals to tolerate higher body heat loads, which are mainly dissipated by drastic increase in BFs at times when solar radiation is minimal or non-existent [34,43,44]. These results coincided with previous investigations, since the lower solar radiation was related to lower BF during the morning.

From a physiological point of view, an increase in BF helps an animal to dissipate excess heat through the respiratory tract, thanks to an increase in the frequency and a decrease in the volume of inspired air. Evaporation in the respiratory tract depends on the volume of air that circulates on the moist surfaces of the same, ventilation is the result of the frequency and depth of breathing. The rhythm of this rapid and shallow breathing (panting) is an indicator of respiratory evaporation [37], which is observed in the two groups during the afternoon. The main physiological mechanisms of thermoregulation activated by ewes under scenarios of high environmental temperatures are an increase in BF and, consequently, an increase in RT [11,45]. The RT values reported in this study are similar to those in the study by Gastelum-Delgado et al. [43], changing only the THI (>80 U). In another investigation, pure-bred Blackbelly ewes and crosses with Dorset were more tolerant to heat than ewes of hair breeds such as Pelibuey, Dorper, Katahdin, or their crosses, as they presented lower RT values under heat stress in tropical environments [46,47].

In ewe production systems with climates classified as hot-humid tropical environments, an average RT of 39.4 °C was reported in Katahdin lambs exposed to heat stress under confinement conditions, which remained within the normal values for ewes and similar to that obtained in Creole ewes fed under grazing [41]; similar characteristics were present in the environmental setting where the present work with Blackbelly ewes was developed, since maximum environmental temperatures of 38.8 °C were recorded during the afternoon. Therefore, the most important parameter to determine the presence of heat stress can be considered to be an increase in RT [41]. When the homeostatic balance is altered, a heat exchange is activated, which is reflected in a change in RT [48]. In this study, the ewes presented changes in their RT by time of day, which could indicate adaptation to THI values higher than the comfort zone [41]. Another physiological parameter that has been recorded under heat stress conditions is an increase in corporal temperature. Animals in heat stress, transport heat from inside the body to the skin through blood flow [41]. The RT showed slightly higher values than those reported for animals in the comfort zone [46,47]. The mean HR was close to the ranges considered acceptable for ewes under tropical environmental conditions (117.8 bpm) [48].

The RM of ewes under comfort conditions varies from two to three movements every two minutes. In the present study, the values recorded were below normal because ewes of hair breeds in heat stress make physiological adjustments by decreasing metabolic activity and feed consumption, simultaneously with an increase in BF [22,34,44]. Few studies have considered the effect of the time of day (morning and evening) on SE, CRT, and RM in ewes. In the present investigation, increases in CRT and RM were observed during the afternoon, when the THI also reached the maximum value. RM is the number of times that the rumen moves in order to mix and process its content, however, when feed consumption is reduced by the effect of heat stress and water consumption is increased, these movements can decrease RM frequency [37].

The CRT is defined as the time required for the return of pale color to normal pink after the application of pressure on the gingiva [31,48]. In unhealthy animals, the CRT can indicate poor circulation causing peripheral tissue perfusion [49]. In the present study, the CRT was greater during the afternoon, regardless of the experimental group. This is probably due to the fact that the ewes are stressed during the non-evaporative phase where the blood is in the periphery.

In the present study, the maximum recorded RT in lambs was 40.6 °C at environmental temperatures greater than 34 °C, indicating a physiological change to counteract the heat stress in the afternoon. Heat stress in ewes is detected through changes in RT and BF. The RT is the physiological constant with the greatest sensitivity to express changes in internal body temperature, considering that ewes are in heat stress when the values of this variable are higher than 39.9 °C [11]. Large variations between the minimum and maximum values of physiological variables (until 39.5 °C for RT and 142 bpm for BF) were clearly observed between the morning and afternoon shifts. Naturally, RT and THI are higher in the afternoon and, consequently, so are the values of other physiological variables [8,11]. In addition, the averages of RT (39.5 °C) and BF (110 bpm) observed in the present study coincided with results (RT = 39.2 °C and BF = 115 bpm) of a previous study carried out in the north of Mexico during the summer season [18].

### 4.3. Skin Temperatures

Vicente-Pérez et al. [46] reported an ST range of 24.4–32.7 °C in different body regions. In the present investigation, ewes exceeded this range, since temperatures of 42.01 ± 6.93 °C were recorded, as was the case in the anatomical area of the haunch. In an investigation by Gómez-Guzmán et al. [47], they reported that a THI value of 79 U affected the temperature of the belly surface, reporting up to 38 °C; the results of this study remained at 36 °C for the same body region, but with a THI value of 76.5 U. This effect was also observed in the haunch, where the author reported 41.11 °C (THI value of 79 U), and in our case, a temperature of 42 °C was recorded in the same region with a THI value of 83.9 U.

The results of this study are in agreement with investigations where the temperatures of different body regions of ewes increase as well as the ambient temperature of the environment [46,47,48]. Finally, the head is one of the best regions to evaluate an increase in temperature in ewes since head temperature is directly correlated with the environmental temperature [48]. In the present study, regardless of the group, head temperature was higher during the afternoon. This evidences the high ambient temperatures that occur in the region of the Costa Grande of Guerrero.

The results of research on heat stress have determined that environmental temperature and relative humidity affect the surface temperature of different body regions of ewes, causing a stress condition that affects both productive and reproductive parameters [47,48].

## 5. Conclusions

Adult Blackbelly ewes in the tropics of Guerrero are capable of dissipating heat under conditions of severe heat stress, making changes in physiological variables to thermoregulate during the heat load period. This allows the effects on breeding stock production in these environmental conditions to be less adverse, and therefore, it is imperative to continue working with this breed due to its high adaptation and resistance to heat stress.

## Figures and Tables

**Figure 1 animals-12-01860-f001:**
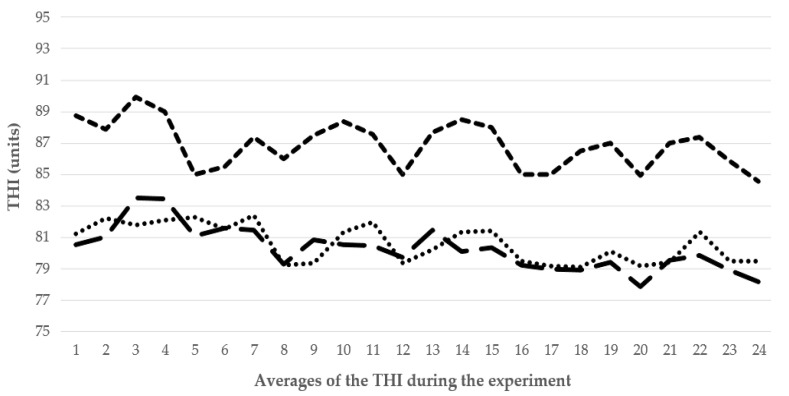
Temperature-humidity index (THI) values during the morning (dotted line), afternoon (line with short dash), and night (line with long dash) of the experiment.

**Table 1 animals-12-01860-t001:** Average weight and body condition (BC) of the experimental groups.

Groups	Variables	Mean ± SD	CV	Min	Max
Adult ewes	Weight (kg)	35.32 ± 1.41	14.96	25.1	44.1
	BC (U)	2.57 ± 0.09	13.44	2	3
Female lambs	Weight (kg)	19.27 ± 1.25	19.59	15.7	25.7
	BC (U)	2.36 ± 0.073	9.33	2.25	2.75

SD, standard deviation; CV, coefficient of variation, minimum and maximum.

**Table 2 animals-12-01860-t002:** Descriptive statistics for climatic variables (mean ± standard deviation (minimum and maximum)).

Variables	Morning			Afternoon			Evening		
	Mean ± DE	Min	Max	Mean ± SD	Min	Max	Mean ± SD	Min	Max
Environmental temperature (°C)	29.3 ± 3.1	24.8	36.8	34.2 ± 2.9	25.8	38.8	28.56 ± 2.06	24.7	36.8
Relative humidity (%)	70 ± 10	50	90	70 ± 10	45	85	78.62 ± 0.07	58	89
Wind speed (km/h)	4.5 ± 1.6	1.8	10.5	6.1 ± 2.1	1.7	19.5	4.22 ± 1.72	1.6	18.8
Solar radiation (W/m^2^)	217.8 ± 256.7	0	886	561.7 ± 346.7	14	1190	17.9 ± 59.66	0	562

SD, standard deviation, CV, coefficient of variation (minimum and maximum)

**Table 3 animals-12-01860-t003:** Physiological constants of heat-stressed ewes during summer in a tropical region.

Groups	Time of the Day	n	Variables	Mean ± SD	CV	Min	Max
		490	BF (bpm)	42.15 ± 18.12	42.99	14	120
	Morning		HR (bpm)	91.99 ± 18.08	19.65	16	192
G1			RT (°C)	38.18 ± 1.68	4.40	28.1	68.5
	Afternoon	490	BF (bpm)	116.73 ± 33.59	28.78	28	196
			HR (bpm)	108.58 ± 20.38	18.77	32	176
			RT (°C)	38.93 ± 0.87	2.24	30.7	40.9
		281	BF (bpm)	36.84 ± 14.97	40.65	16	96
	Morning		HR (bpm)	95.39 ± 20.42	21.41	32	188
G2			RT (°C)	38.54 ± 0.54	1.401	34.1	40.7
		281	BF (bpm)	113.66 ± 34.51	30.36	32	192
	Afternoon		HR (bpm)	110.43 ± 22.33	20.22	31.1	168
			RT (°C)	39.26 ± 0.47	1.21	34.2	40.6

n, number of observations; SD, standard deviation; CV, coefficient of variation (minimum and maximum).

**Table 4 animals-12-01860-t004:** Effect of time of day (morning and evening) on skin elasticity (SE), capillary return time (CRT), and rumen movements (RM) in ewes.

Variables	Morning		*p* Value	Afternoon		*p* Value
	G1	G2		G1	G2	
SE (min)	1.27 ± 0.48	1.21 ± 0.43	0.579	1.19 ± 0.41	1.23 ± 0.42	0.56
CRT (min)	1.14 ± 0.38	1.18 ± 0.40	0.006	1.15 ± 0.41	1.24 ± 0.44	0.005
RM (pdm)	1.20 ± 0.40	1.09 ± 0.264	<0.0001	1.39 ± 0.48	1.23 ± 0.42	<0.0001

Mean ± standard deviation. A highly significant difference is considered at *p*-value < 0.001.

**Table 5 animals-12-01860-t005:** Skin temperatures of heat-stressed G1 and G2 ewes during summer in a tropical region.

Body Area	Morning (°C)		*p* Value	Afternoon (°C)		*p* Value
	G1	G2		G1	G2	
Head	31.23 ± 17.611	29.72 ± 2.131	<0.0001	38.48 ± 4.04	37.65 ± 3.58	<0.0001
Scapula	29.79 ± 1.71	29.83 ± 1.91	<0.0001	36.61 ± 3.01	36.44 ± 3.13	<0.0001
RPF	28.80 ± 1.95	28.68 ± 2.05	<0.0001	39.31 ± 5.05	38.25 ± 4.19	<0.0001
Rump	27.75 ± 1.70	27.68 ± 1.66	<0.0001	42.01 ± 6.93	40.34 ± 5.88	<0.0001
Leg	28.87 ± 1.67	28.59 ± 1.99	<0.0001	36.95 ± 3.22	36.50 ± 3.21	<0.0001
Stomach	29.56 ± 2.22	28.84 ± 2.18	<0.0001	35.90 ± 2.45	36.01 ± 2.91	<0.0001

Mean ± standard deviation. Right paralumbar fossa (RPF). A highly significant difference is considered at *p*-value < 0.001.

## Data Availability

Not applicable.

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
