# Peer review of "Thermoregulatory Response of Blackbelly Adult Ewes and Female Lambs during the Summer under Tropical Conditions in Southern Mexico"

_animals, 2022, doi:10.3390/ani12141860_

Round 1

Reviewer 1 Report

 Thermoregulatory Response of Blackbelly Lambs and Ewes

There are major fundamental problems with the some of the most important variables used in this study that make it not acceptable for publication. Similarly, there are some major problems with the statistical analysis that necessitate rejection of the paper.  I gave up on reviewing the paper at about line 186 because it is just too flawed to continue reviewing.  

23          to  “increasing heat stress (HS)”.  There was always heat stress to some degree. 

26-28    Consider rewording this, and why are you now using “females”?

29          Use “challenges” in stead to problems.  It would also be good to list the other items you measured.    

32          Define SE or correct the wording.

35          Tell us what was higher or lower. 

39          List the physiological variables you measured so this is more specific.

45-51    Good intro.

67-69    Their genes cause the mechanisms you listed, not “in addition”.

83          Need to define BF.

88          Technology is vague, give the specifics you talked about to make it clearer.

100        Reword, without the brackets. 

119        Something is wrong.  Do you mean HR?

133        HR is highly responsive to someone entering a pen, restraining an animal and using a stethoscope.  How do we know that ewes vs lambs did not just react differently to this procedure? This is a major problem. Also, the first animal in the group caught is very likely to have a HR much different than the last animal caught in the group.  More detail on this problem is absolutely necessary for this paper to be publishable.  

 136       Thermography is very much influenced by radiant heat - sunshine.  One cannot merely take shots of an animal without considering whether it was standing in the sunshine, for how long, what part of the animal were exposed to the sun, etc.                                                         

143        Oh no,  the animals were in an open pen, subjected to direct sun light, but shade was also available. This makes things much more complicated.     

146        “the ewes remained grazing”, but where were the lambs??  Lying in the shade??

186        Table 3 shows that you were using every reading as an experimental unit, which is wrong.          

Reviewer 2 Report

Additional information:

line 30: (and 32 and 35), lambs were also ewes? Please add the sex of lambs!!

line 47: rather animal production!

line 50: please add full words of HS (not enough is ad in abstract)!

lines 54-55: please clarify the „ewes”! It seems, you think mother ewes! ewes call the young lambs, too!

line 89: please add your hypothesis!

line 104: please add age of ewes!

line 106: please add the description of body condition in the Materials and Methods section!

lines 129-131: please add the list: skin temperature!

lines 132-140: how measured the skin elasticity, capillary return time and ruminal movements? Presently, no information was available on these traits in this section!

line 144: length of feeders was same for animals in both groups?

Table 2: relative humidity maximum values are incorrect!

lines 172-175: max unit was 89? Although, in Figure 1 found THI=90! THWRI is not find in Figure 1 (see line 175)!!

Table 3: please reedit similar to Table 4 and 5!

Table 4: please correct the table (underlined the SE trait)!

Table 4 and 5: Significance indications are incorrect! P columns are need: between morning G1-G2; between afternoon G1-G2; and between morning and evening!  Or use part of the day groups (2 groups) and treatment groups (also 2 groups) with P values!

line 249: ITHA? Please add full phrase!

line 274: TR or RT?

line 290: HT: what does it mean?

line 294: please discussed the CRT!

line 329: Is the average value of RT 3 0C? FR?

lines 320-331: please move to 3.2 subsection!

Conclusion: see line 290! Rather physiological traits/variables than constants! lines 340-341: more focused on own results! Conclusion should not simply repeat the main results but try to put these results in a broader perspective. Please rewrite in a context of heat stress and response to this!

Author Response

Comments and Suggestions for Authors

Additional information:

line 30: (and 32 and 35), lambs were also ewes? Please add the sex of lambs!!: Observation accepted and applied to the manuscript

line 47: rather animal production!: Observation accepted and applied to the manuscript.

line 50: please add full words of HS (not enough is ad in abstract)!: The words heat stress are incorporated in full throughout the document by removing the abbreviation HS

lines 54-55: please clarify the „ewes”! It seems, you think mother ewes! ewes call the young lambs, too! The clarification is made, throughout the document "adult sheep" and "female lambs" are specified to identify groups.

line 89: please add your hypothesis!: Hypothesis added to the paragraph

line 104: please add age of ewes!: This data is specified in the paragraph

line 106: please add the description of body condition in the Materials and Methods section!: The description of body condition in the animals and treatment paragraph in subsection 2.2 is clarified.

lines 129-131: please add the list: skin temperature!: Skin temperatures have been added to the list of variables, in the same way it has been described how they were recorded.

lines 132-140: how measured the skin elasticity, capillary return time and ruminal movements? Presently, no information was available on these traits in this section! Specific information where heat stress is evaluated in sheep does not currently exist. However, we made the measurements as commonly established by the veterinary clinic on the physical examination method by means of general inspection, palpation, percussion and auscultation to detect clinical signs in animals. We observed that it was interesting to carry out the evaluation of these variables in sheep under heat stress since they are compromised. Some published works are presented where these practices are supported.

Tagesu A (2018) Physical Examination. Int J Vet Sci Res s1: 007-013. DOI: http://dx.doi.org/10.17352/ijvsr.s1.102

Jackson P, Cockcroft P (2002) Clinical Examination of Farm Animals. Blackwell Science, UK.

Duguma A (2016). Practical Manual on Veterinary Clinical Diagnostic Approach. J Vet Sci Technol 7: 337. doi:14.4172/2157-7579.1000337

Radostits O.M. Gay C.C: Blood D.C. Hinchcliff K.W. (2007). Veterinary medicine: a textbook of the diseases of cattle, horses, sheep, pigs, and goats. 10th ed. Philadelphia: W.B. Saunders, 2180 p.

line 144: length of feeders was same for animals in both groups? The length of the feeders was the same, in fact, all the animals received the same handling and were not separated. They came from grazing and were locked in the rest pen.

Table 2: relative humidity maximum values are incorrect! Correction is made in the minimum and maximum values of temperature and humidity in Table 2

lines 172-175: max unit was 89? Although, in Figure 1 found THI=90! THWRI is not find in Figure 1 (see line 175)!!, The correction was made in the paragraph and it is specified that Figure 1 is only for the ITH values, the change has been made for the observation.

Table 3: please reedit similar to Table 4 and 5! The arrangement of the tables has been made in the indicated format

Table 4: please correct the table (underlined the SE trait)! The change has been made

Table 4 and 5: Significance indications are incorrect! P columns are need: between morning G1-G2; between afternoon G1-G2; and between morning and evening!  Or use part of the day groups (2 groups) and treatment groups (also 2 groups) with P values!, The changes were made of the observations made with respect to Tables 4 and 5.

line 249: ITHA? Please add full phrase!, The indicated abbreviation was changed and incorporated in the text.

line 274: TR or RT? Changed to RT to specify rectal temperature

line 290: HT: what does it mean? The specification for rectal temperature has been made.

line 294: please discussed the CRT! CRT was discussed as suggested by the reviewer.

line 329: Is the average value of RT 3 0C? FR? The observation was attended to and applied to the manuscript.

lines 320-331: please move to 3.2 subsection! The suggested paragraph was changed to subsection 3.2.

Conclusion: see line 290! Rather physiological traits/variables than constants! lines 340-341: more focused on own results! Conclusion should not simply repeat the main results but try to put these results in a broader perspective. Please rewrite in a context of heat stress and response to this! The observation was accepted and the conclusion was changed 

Round 2

Reviewer 1 Report

This still has the same fundamental errors in the design of the study that make this unpublishable.  There are also still careless errors in the English and wording.

Reviewer 2 Report

The presentation of the results, the scientific soundness and the conclusions of this manuscript were improved by authors, so I recommend this manuscript for publishing in the Animals journal.